# Association between physical activity levels in mid-life with physical activity in old age: a 20-year tracking study in a prospective cohort

Daniel Aggio,[1,2] Olia Papacosta,[1] Lucy Lennon,[1] Peter Whincup,[3]
Goya Wannamethee,[1] Barbara J Jefferis[1,2]

► Prepublication history and additional material are available. To view these files please visit the journal online (http://dx.doi.org/10.1136/bmjopen-2017-017378).

[1]UCL Department of Primary Care and Population Health, UCL Medical School, London, UK
[2]UCL PA Research Group, London, UK
[3]Population Health Research Institute, St George's University of London, London, UK

**Correspondence to**
Dr Daniel Aggio;
d.aggio@ucl.ac.uk,
danielaggio1@hotmail.co.uk

## ABSTRACT

**Objectives** This study aims to examine the tracking and predictability of physical activity in old age from overall physical activity and participation in sport, recreational activity and walking in mid-life.

**Design** Prospective population-based cohort study.

**Setting** British Regional Heart Study participants recruited from primary care centres in the UK in 1978–1980.

**Participants and outcome measures** Men (n=3413) self-reported their physical activity at baseline, 12, 16 and 20-year follow-ups and were categorised as inactive or active and having high or low participation in sport, walking and recreational activities. Tracking was assessed using kappa statistics and random effects models. Logistic regression estimated the odds of being active at 20-year follow-up according to physical activity participation in mid-life.

**Results** Among 3413 men (mean age at baseline 48.6±5.4 years) with complete data, tracking of overall physical activity was moderate (kappa: 0.23–0.26). Tracking was higher for sports participation (kappa: 0.35–0.38) compared with recreational activity (kappa: 0.16–0.24) and walking (kappa: 0.11–0.15). Intraclass correlation coefficients demonstrated similar levels of stability and only marginally weakened after controlling for covariates. Compared with inactive men, being active at baseline was associated with greater odds of being active at 20-year follow-up (OR 2.7, 95% CI 2.4 to 3.2) after adjusting for sociodemographic, health and lifestyle variables. Playing sport in mid-life was more strongly associated with being active at 20-year follow-up than other domains, particularly when sport participation began earlier in life.

**Conclusion** Being physically active in mid-life increases the odds of being active in old age. Promoting physical activity in later life might be best achieved by promoting sport participation earlier in the life course.

## INTRODUCTION

Prospective epidemiological studies have shown that physical activity (PA) in mid-life and old age is associated with numerous health benefits, including reductions in cardiovascular disease events and mortality.[1–3]

## Strengths and limitations of this study

► This study investigates the tracking of overall and specific domains of physical activity during the transition to old age over 20 years, an understudied period of the life course.
► Very few studies have investigated the tracking of specific domains of physical activity during this period.
► Our results may not be generalisable to women and non-white ethnic groups.

Taking up PA in later life may reduce the risk of adverse health outcomes, but maintaining a physically active lifestyle throughout the life course may provide optimal health benefits.[4–6] The transition from mid-life to old age typically coincides with major life events (eg, retirement) and therefore may be an important window when both the volume and type of PA are likely to change. Knowledge on the stability, or tracking, of PA during this transition is very limited. The tracking of a behaviour over time can be determined by (1) its stability of overtime, typically estimated using correlations between repeated measures or (2) the predictability of later measures from previous ones.[7] Past exercise behaviour is a consistent predictor of current PA levels[8]; however, few studies have examined the predictability of PA in old age from PA measures in mid-life. Understanding tracking of PA during this transition may help inform interventions aiming to promote or maintain activity levels from mid-life to old age.

There is a large body of research on the tracking of PA from childhood, but few studies have extended over prolonged periods in adults.[9] Current evidence suggests low to moderate tracking of PA throughout the life course.[9–11] Studies tracking PA in

youth have shown that sport participation in early life tracks more strongly[9] and is a stronger predictor of activity levels in adulthood (age 42 years) compared with other domains of activity such as outdoor play.[12] However, tracking studies in adults have rarely distinguished between the types of PA. Thus, it remains unknown what types of activity in mid-life are more likely to predict PA in old age. The limited evidence in older adults has suggested some domains of PA are more liable to change (eg, indoor activities) than others (eg, outdoor and leisure activities)[13] and thus may be easier to modify. Further studies have investigated the predictability of activity levels in early old age according to PA in early adulthood. For example, one study showed that being moderately active in young adulthood (mean age 35 years) increased the odds of being active 28 years later by more than three times.[14] Another study showed that sport participation in healthy young men (aged 25 years) strongly predicted PA 50 years later.[15] However, this study was retrospective in nature and may not be generalisable to less healthy populations.

Overall, very few tracking studies have extended into old age. Furthermore, the predictability of PA in later life from participation in specific types of activity in mid-life remains unknown. Thus, we aimed to estimate the tracking of overall and specific domains of PA from mid-life to old age and the predictability of PA levels in old age from (1) overall PA and (2) PA domains in mid-life.

## METHODS

### Participants

Data were drawn from the British Regional Heart Study, an ongoing prospective cohort study involving 7735 men (response rate=78%) from 24 towns in Great Britain.[16] Men were recruited from primary care practices and were first examined in 1978–1980 aged 40–59 years and were followed up after 12, 16 and 20 years. Response rates for surviving cohort members were 91% (n=5925), 88% (n=5263) and 77% (n=4252) at 12, 16 and 20-year follow-ups, respectively. Men completed a lifestyle and medical history questionnaire at the time of the examination (baseline and 20-year follow-up) or by post (12 and 16-year follow-ups). Participants provided informed written consent to the investigation. Ethical approval was obtained from the National Research Ethics Service Committee London.

### Measures

#### Self-reported PA

At all waves, participants reported their usual PA levels. Questions referred to time spent on all forms of walking, time spent on recreational activities (such as recreational walking, gardening, chores, do-it-yourself (DIY)) and how frequently they participate in sport/exercise. Responses to each domain of PA were scored based on the intensity and frequency of the activity.[17 18] For example, making no journeys by foot was scored as 0 and >90 min/weekday was

scored as 5. Scores were also heavily weighted for vigorous activities. For example, playing sport four to seven times a month was given a score of 8. Scores for each domain were summed together to give a total PA score. The original scoring system has been reported in detail elsewhere.[19] The total PA score was then used to classify activity levels as inactive, occasional, light, moderate, moderately vigorous or vigorous. These PA scores have previously been validated against resting heart rate[19] and accelerometer-measured PA.[20] Results from the validation studies revealed a strong inverse relationship between PA and ECG-measured resting heart rate (p<0.001) and a strong positive association with accelerometer-measured moderate-to-vigorous PA (r=0.49, p<0.001).[19 20] For the purposes of this study, the categories were grouped into active or inactive (inactive and occasional groups were classified as inactive). Responses to individual questions were also used to classify participation in specific domains of activity. Men were classified as having high or low sport/exercise participation (no sport/exercise participation was classified as low), high or low walking (low walking was classified as ≤20 min/day) and high or low recreational activity (low recreational activity was defined as being similar or less active than someone who spends 2 hours on most days on recreational activities). Men who reported participating in sport also retrospectively disclosed how many years they had been involved in that activity, from which men were classified as participating in sport for ≤4 years, 5–11 years, 12–24 years and ≥25 years.

### Covariates

Participants self-reported their age at baseline; social class, which was derived from their longest held occupation[21] and categorised as manual or non-manual; and cigarette smoking habits, classified as current or ex-smokers and never smokers. Nurses measured participant's height and weight, which was used to derive body mass index (BMI). Men were then categorised as overweight or obese (BMI: ≥25.0 kg/m$^2$) or healthy weight (BMI: <25.0 kg/m$^2$).

### Statistical analysis

Descriptive statistics were used to report sample characteristics at baseline and the proportion of men active/high participation at each wave. McNemar's $\chi^2$ test was used to determine whether the proportion reporting being physically active changed between time points. Cohen's kappa was used to assess the observed agreement compared with the expected agreement. We followed suggestions by Munoz and Bangdiwala for interpretation of K coefficients: <0.00 indicates poor agreement, 0.00–0.20 fair agreement, 0.21–0.45 moderate agreement, 0.46–0.75 substantial agreement and 0.76–1.0 indicates near perfect agreement.[22] Kappa statistics vary in magnitude depending on how the outcome measure is categorised. To be consistent across all our measures we decided to perform analyses using binary variables. Random effects models were also used to calculate intraclass correlation

coefficients (ICCs), providing an indicator of tracking using data from all assessments while also controlling for covariates. In a supplementary analysis, we stratified our sample according to changes in employment status as we hypothesised that the timing of retirement may affect the stability of PA. We categorised men as no change in employment status (representing continuous employment/seeking employment and continuously retired) or retiring (ie, retired between baseline and the respective follow-up) and presented kappa statistics separately. Finally, we used logistic regression to estimate the OR for being active compared with being inactive at 20-year follow-up according to (1) overall activity levels at baseline, (2) engagement in specific domains of PA at baseline and (3) duration of sports participation. Tests for linear trend were also conducted by entering sports duration as a continuous variable into regression models. Initial models were adjusted for age, entered as a continuous variable (model 1) and then for BMI, social class and smoking status (categorical) (model 2). In analyses using just baseline activity levels as predictors of activity (ie, not sports duration) at 20-year follow-up, we also introduced a third model including all PA variables to identify the strongest predictor of activity 20 years later while also accounting for participation in other types of PA.

## RESULTS

Up to 7735 men responded to the baseline survey. Men who died during follow-up (26.4%, n=2041), those with missing PA data for other reasons (29.4%, n=2272) at one or more examination between baseline and 20-year follow-up and those with missing covariate data (0.1%, n=9) were excluded from analyses, leaving 3413 for analyses. Compared with men in the analytical sample, men excluded from the analyses were significantly older

(baseline age, 48.6 vs 51.5 years, p<0.001), had a higher BMI (baseline BMI, 25.3 vs 25.7, p<0.001) and were more likely to be inactive at baseline (proportion inactive at baseline, 55.5% vs 66.1%, p<0.001). A larger sample was included in the random effects models, as men were only excluded if they did not provide PA measures on at least two assessments and have valid covariate data.

Table 1 displays sample characteristics and the proportion of men who were physically active and who participated in PA domains at each time point. Between baseline and 12-year follow-up, the number of men classified as active increased from 66.1% to 71.0% (p<0.001) and then dropped significantly to 63.7% and 66.9% (p<0.001) at 16 and 20-year follow-ups, respectively. The proportion of men classified as active declined more rapidly thereafter, with 57.3% of men classified as active at 30-year follow-up (data not shown). The proportion of men reporting high levels of walking increased from 26.9% at baseline to 61.5% at 20-year follow-up (p<0.001). There were also steep declines over the 20-year follow-up in recreational activity, with 56.0% of men reporting high levels of recreational activity at baseline and 40.2% at 20-year follow-up.

Table 2 presents kappa statistics and ICC for PA variables. Kappa statistics for overall PA ranged from 0.23 to 0.26 between baseline and subsequent time points, but were highest for sports participation (0.35–0.38) and lowest for walking (0.11–0.15). Kappa statistics were generally higher for shorter follow-up periods. In random effects models, ICCs were consistent with the kappa statistics and were only marginally weakened after controlling for covariates. In a supplementary analysis, we present kappa statistics according to employment status. Overall stability of total PA was similar between men who reported no change in working status and men who

**Table 1**  Sample characteristics and physical activity levels at baseline, 12, 16 and 20-year follow-ups, n=3413

|  | Baseline | 12 years | 16 years | 20 years |
|---|---|---|---|---|
| Age (years, mean±SD) | 48.6±5.4 | 62.2±5.4 | 66.2±5.4 | 68.5±5.4 |
| Overweight/obese (%, n)* | 52.2 (1783) | | | |
| Current smoker (%, n)* | 30.6 (1043) | | | |
| Manual occupation (%, n)* | 50.2 (1713) | | | |
| Physically active† (%, n) | 66.1 (2257) | 71.0 (2422) | 63.7 (2173) | 66.9 (2284) |
| High sport participation‡ (%, n) | 47.7 (1627) | 45.3 (1532) | 44.6 (1493) | 49.2 (1663) |
| High recreational activity§ (%, n) | 56.0 (1912) | 58.4 (1994) | 41.2 (1407) | 40.2 (1372) |
| High walking¶ (%, n) | 26.9 (918) | 51.6 (1754) | 50.9 (1735) | 61.5 (2097) |

Data presented are for participants with a valid physical activity score at all four time points (n=3413). Data on walking were missing for an additional 15 participants at 12-year follow-up, 3 participants at 16-year follow-up and 1 participant at 20-year follow-up. Data on sport participation were missing for 33 participants at 12-year follow-up, 68 participants at 16-year follow-up and 34 participants at 20-year follow-up.
*Data on BMI, smoking status and occupational class were used at baseline only.
†Physically active was classified as reporting at least light activity.
‡High sport was classified as reporting at least occasional participation (less than once a month).
§High recreational activity was classified as >2 hours/day on recreational activities.
¶High walking was classified as >20 min/day.

**Table 2** Stability of physical activity variables over time, n=3413

| | Waves 1–2 | Waves 1–3 | Waves 1–4 | Random effects models | |
| | | | | Univariate | Multivariate* |
| | Kappa | Kappa | Kappa | ICC (95% CI) | ICC (95% CI) |
|---|---|---|---|---|---|
| Physically active | 0.26 | 0.23 | 0.24 | 0.46 (0.43 to 0.48) | 0.44 (0.41 to 0.46) |
| Sport participation | 0.38 | 0.35 | 0.35 | 0.65 (0.63 to 0.67) | 0.61 (0.59 to 0.63) |
| Recreational activity | 0.24 | 0.19 | 0.16 | 0.38 (0.36 to 0.40) | 0.36 (0.34 to 0.39) |
| Walking | 0.15 | 0.11 | 0.12 | 0.32 (0.30 to 0.35) | 0.32 (0.30 to 0.34) |

Kappa statistics are presented for participants with a valid physical activity score at all four time points (n=3413). Data on walking were missing for an additional 15 participants at 12-year follow-up, 3 participants at 16-year follow-up and 1 participant at 20-year follow-up. Data on sport participation were missing for 33 participants at 12-year follow-up, 68 participants at 16-year follow-up and 34 participants at 20-year follow-up. Random effects models included men with at least two assessments for each domain accompanied by valid covariate data (physical activity: n=5962; sport participation: n=6122; recreational activity: n=6093; walking: n=6040). ICC from random effects models.
*Adjusted for age, body mass index, social class and smoking status at baseline.
ICC, intraclass correlation coefficients.

retired between baseline and subsequent follow-ups (see online supplementary table 1). However, a higher proportion of men who were retiring increased their total activity between baseline and subsequent follow-ups compared with men who reported no change in their working status (eg, 21.3% vs 15.7% of men increased their total activity levels between wave 1 and wave 2 in the retiring group and the no change group, respectively (data not shown)). Similarly, the overall stability of sport participation was comparable between men who reported no change in working status and retiring men, but the retiring group contained a higher proportion of men who increased their sport participation (eg, 15.8% vs 12.6% of men increased their sports participation between wave 1 and wave 2 in the retiring group and the no change group, respectively (data not shown)). Stability of recreational activity was markedly lower in men retiring between baseline and wave 4 compared with men who reported no change in their working status during the same period. This was largely explained by a higher proportion of retiring men reporting a decrease in recreational activity compared with men reporting no change in work status (eg, 30.4% vs 24.7% of men reported a decrease in recreational activity between wave 1 and wave 4 in the retiring group and the no change group, respectively). There were also some clear differences in the stability of walking activity between men who reported no change in working status and retiring men. This was largely explained by a higher proportion of retiring men reporting an increase in walking activity compared with men with no change in working status (eg, 39.6% vs 28.9% of men reported an increase in walking activity between wave 1 and wave 2 in the retiring group and the no change group, respectively).

Compared with inactive men, being physically active at baseline was associated with greater odds (OR 2.7, 95% CI 2.4 to 3.2) of being physically active at 20-year follow-up after adjusting for age, social class, BMI and smoking status at baseline (table 3). ORs for being active at 20-year follow-up were similarly raised for men who played sport

at baseline after adjustments (OR 2.7, 95% CI 2.3 to 3.1). High participation in walking and recreational activity at baseline were also associated with greater odds of being active at 20-year follow-up (OR 1.4–1.6). In the final model including all PA domains, sport participation at baseline remained the strongest predictor of being active at 20-year follow-up.

Table 4 shows the odds of being active at 20-year follow-up according to duration of sports participation from baseline. This sample size was lower than the main analytical sample as 27.3% (n=444) of men who played sport did not report duration of participation. Longer duration of sports participation was associated with increased odds of being active at 20-year follow-up. Compared with those who were not participating in sport at baseline, taking part in sport for 25 years or more was most strongly associated with being active at 20-year follow-up (OR 4.8, 95% CI 3.4 to 6.8). However, even taking up sport recently (≤4 years) was associated with increased odds of being active at 20-year follow-up (OR 2.3, 95% CI 1.7 to 3.2).

## DISCUSSION

This study investigated the tracking of overall and specific domains of PA from mid-life to old age and the predictability of PA in old age from overall PA and participation in sport, recreational activity and walking in mid-life. Agreement between overall PA levels at baseline and subsequent measures at 12, 16 and 20-year follow-ups suggested moderate levels of tracking, with stronger tracking evident for sport participation compared with other domains. Importantly, however, prevalence of walking >20 min/day increased from around a third to approximately two-thirds by 20-year follow-up. Comparisons with previous studies are challenging given the various measures, cut-off points and time frames studied, but our findings appear to be in agreement with previous studies in similar age groups,[9 23 24] but over a longer time span. As expected, tracking coefficients tended to be higher for

**Table 3** Odds of being active at 20-year follow-up according to activity levels at baseline, n=3413

| | N | Model 1 OR (95% CI) | Model 2 OR (95% CI) | Model 3 OR (95% CI) |
|---|---|---|---|---|
| **Physical activity** | | | | |
| Inactive | 1156 | 1.0 | 1.0 | — |
| Active | 2257 | 2.9 (2.5 to 3.3) | 2.7 (2.4 to 3.2) | — |
| **Sport** | | | | |
| Low | 1786 | 1.0 | 1.0 | 1.0 |
| High | 1627 | 2.9 (2.5 to 3.4) | 2.8 (2.4 to 3.3) | 2.7 (2.3 to 3.1) |
| **Recreational activity** | | | | |
| Low | 1501 | 1.0 | 1.0 | 1.0 |
| High | 1912 | 1.9 (1.6 to 2.2) | 1.8 (1.6 to 2.1) | 1.6 (1.4 to 1.9) |
| **Walking** | | | | |
| Low | 2495 | 1.0 | 1.0 | 1.0 |
| High | 918 | 1.5 (1.3 to 1.8) | 1.5 (1.3 to 1.8) | 1.4 (1.2 to 1.7) |

Model 1 adjusted for baseline age. Model 2 additionally adjusted for social class, BMI and smoking status at baseline. Model 3 mutually adjusted for all domains of activity respectively.

the shorter follow-up periods. Although the proportion of men categorised as active fluctuated over the 20-year follow-up period, we did not observe the decline over time that one might expect with the advancing age of the sample. However, when we extended our follow-up to 30 years when men were aged 70–89, a notable decline was observed. This is consistent with cross-sectional data from the Health Survey for England,[25] which presented similar proportions of men meeting PA recommendations at ages 55–64 and 65–74 years, followed by a decline in men >75 years old.

This is the first study that we are aware of to examine tracking of specific domains of activity from mid-life to old age and the predictability of PA in old age from PA domains in mid-life. Sports participation was the most

**Table 4** Odds of being active at 20-year follow-up according to duration of sport participation at baseline (n=2969)

| | N | Model 1 OR (95% CI) | Model 2 OR (95% CI) |
|---|---|---|---|
| **Sports participation duration** | | | |
| Not participating at baseline | 1786 | 1.0 | 1.0 |
| ≤4 years | 262 | 2.4 (1.8 to 3.2) | 2.3 (1.7 to 3.2) |
| 5–11 years | 331 | 3.0 (2.2 to 4.0) | 2.9 (2.2 to 3.9) |
| 12–24 years | 290 | 4.4 (3.1 to 6.2) | 4.3 (3.1 to 6.0) |
| ≥25 years | 300 | 5.0 (3.6 to 7.1) | 4.8 (3.4 to 6.8) |
| p Value for trend* | | <0.001 | <0.001 |

Model 1 adjusted for baseline age. Model 2 additionally adjusted for social class, body mass index and smoking status at baseline. *Tests for linear trend were conducted by entering sports duration as a continuous variable into regression models.

stable domain with moderate agreement between baseline and subsequent time points. Tracking was fair for walking owing to a high proportion increasing their walking activity (eg, 39.6% of retired men increased from low to high walking between baseline and 12-year follow-up). Tracking of recreational activity was fair to moderate despite a large proportion decreasing their recreational activity by 20-year follow-up (eg, 30.5% of retired men decreased from high to low recreational activity between baseline and 12-year follow-up). Tracking indicators from random effects models provided comparable estimates of tracking, while using all available measurements and controlling for factors that may influence tracking. The differential changes over time between domains of activity may reflect the changing opportunities and functional abilities associated with ageing. Our supplementary analyses suggest that retirement may be a period when PA behaviours are more sensitive to change. Increased free time during retirement may possibly explain the observed steep increase in walking and lower levels of stability during this transition, whereas declines in physical function and onset of chronic disease may explain reductions in the more strenuous activities related to recreational activity, such as DIY and gardening. Given that our measure of recreational activity consisted of several activities including recreational walking, the observed increases in walking may have masked an even steeper decline in other recreational activities.

PA during mid-life was associated with increased odds of being active in old age. Comparable results were found in a study by Morseth *et al*,[14] who found that Norwegian men and women who were active in mid-life (aged 20–54 at baseline) were 5–13 times more likely to remain non-sedentary 28 years later. Although similar, our findings come from a sample of older British men all of whom would have transitioned to old age over the

20-year period and would take part in very different activities to their Norwegian counterparts. Sport participation in mid-life predicted PA in old age more strongly than walking and recreational activity. This is consistent with previous tracking studies from childhood to adulthood that have also shown that sporting activity tracks more strongly[9] and is a stronger predictor of PA later in life than other domains of activity[12 26]; however, this is the first study that we are aware of to demonstrate similar findings during the transition to old age. There may be a number of reasons why participation in sport in mid-life more strongly predicts activity in older age than other types of activity. One possibility is that people's enjoyment of sport may be more likely to persist into old age than preferences for other types of activity. Sport participation in mid-life may help maintain physical function and PA self-efficacy in later life, increasing psychological and physical readiness for PA in old age. Stronger levels of tracking may also suggest that participation in sport is less modifiable than other domains. By contrast, lower levels of tracking for walking, predominantly caused by large increases in uptake, suggest that walking may be easier to adopt in older adults, particularly those with functional limitations. Thus, improving our understanding on how to promote this domain of PA is important for future research. We also found that earlier engagement in sport more strongly predicted PA in old age. Engagement in sport by early adulthood may be crucial for establishing a lifelong habit for sport participation and for developing important motor skills. Thus, earlier engagement may result in improved capability to maintain PA and sport in older age. Although earlier sport participation appears favourable, even taking up sport in mid-life more than doubled the odds of being active in old age compared with adults not taking part in sport in mid-life. Encouraging sport participation early in the life course may be most effective for promoting lifelong PA but even interventions promoting uptake in middle-aged adults may be successful.

The main strengths of this study are its large sample size and long follow-up, encompassing the transition from mid-life to old age, an understudied period in PA tracking studies. Our study is limited by the use of self-reported assessment of PA, which may have been prone to bias. Nevertheless, the questionnaire was validated at baseline against resting heart rate[19] and more recently against accelerometer-measured PA.[20] Despite this, we are unable to validate responses to the question on duration of sport participation, although studies in men of a comparable age have used similar questions with longer recall periods.[15] Use of the same questionnaire at successive waves should result in comparable results between waves. Self-reports also allowed us to examine how specific types of PA track, which may provide useful insight for intervention strategy. Another limitation is that our findings may not be generalisable to women and non-white ethnic groups, who are not represented in this study sample. Furthermore, men who were lost to follow-up were more likely to be overweight or obese and were generally less active than men with complete data. This attrition may have led to an overestimation of PA levels and the strength of tracking. PA may be more liable to change in men who were lost to follow-up, possibly as a result of an increased risk of developing chronic health conditions.[27] Random effects models which provide estimates of tracking using all available data, while also accounting for factors that may influence tracking strength, may have alleviated, at least in part, the bias caused by this attrition.

## CONCLUSION

In conclusion, PA tracks moderately from mid-life to old age. Being active in mid-life was associated with increased odds of being active in old age. Playing sport in mid-life was more strongly associated with PA in old age than other domains of PA. Encouraging early and sustained sports participation into mid-life may be effective for promoting PA in old age, but increased opportunities to take up other forms of activity, such as walking, may also be crucial as people transition into old age.

**Contributors** SGW and PW designed and conceived the study. DA analysed and interpreted the data and drafted the initial manuscript. LL collected the data. OP generated the database. BJJ and SGW interpreted the data and revised the manuscript. OP, SGW, PW, LL, BJJ and DA approved the final manuscript.

**Funding** DA is funded by a British Heart Foundation PhD studentship (FS/15/70/32044). This research was also supported by an NIHR Post-Doctoral Fellowship awarded to BJJ (2010-03-023) and by a British Heart Foundation project grant (PG/13/86/30546) to BJJ. The British Regional Heart study is supported by a British Heart Foundation grant (RG/13/16/30528).

**Competing interests** None declared.

**Ethics approval** Participants provided informed written consent to the investigation. Ethical approval was obtained from the National Research Ethics Service (NRES) Committee London.

**Provenance and peer review** Not commissioned; externally peer reviewed.

**Data sharing statement** Data are not publically available, but applications for data sharing can be made. For enquiries please contact Lucy Lennon (l.lennon@ucl.ac.uk).

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
