## [Reviewer comments · BMJ Open]

ARTICLE DETAILS

TITLE (PROVISIONAL)	Association between physical activity levels in midlife with physical activity in old age: A 20-year tracking study in a prospective cohort
AUTHORS	Aggio, Daniel; Papacosta, Olia; Lennon, Lucy; Whincup, Peter; Wannamethee, Goya; Jefferis, Barbara

VERSION 1 - REVIEW

REVIEWER	Verity Cleland University of Tasmania, Australia
REVIEW RETURNED	23-May-2017

GENERAL COMMENTS	A well and clearly-written paper addressing a research gap of interest – that is, how well does physical activity track from middle age into old age. The study uses a well-established British cohort of men recruited through general practices around 1980, so results won't be generalizable to women or non-white populations, which is appropriately acknowledged. The study found moderate levels of tracking, and provides important data that is not readily available elsewhere. The authors mention in the Abstract that a limitation is the use of a self-report measure of physical activity which may have been prone to recall bias. This measure has been validated against heart rate and respiration, and more recently accelerometry, with acceptable properties. I think in a tracking study, as long as the recall bias is the same at each time point (would we expect people to recall differently at different time points/ages? I don't know) this should not impact on the findings in a significant way. It would only matter if people were recalling physical activity differently at different time points. The authors acknowledge this in the discussion, so I don't think it is warranted here as it detracts from the findings. Measures: the authors of this paper collapsed the physical activity variables into mostly binary categories. I wonder what the justification for this was? Results, page 8, first paragraph: It is reported that those who dropped out of the sample were older, had higher BMIs and were more likely to be inactive at baseline. It would be good if the authors could discuss the potential impact of this attrition on the findings. I acknowledge some will be included in the random effects modelling (those with two assessments) but these people again may be different to those with only one assessment or more than two assessments. Table 1: Not clear why data for overweight/obese, smoking and occupation are absent from all but one assessment? Maybe a
--

	footnote to explain? Table 1: I was surprised by how active this population is, especially at the 20-year follow-up when some participants would be nearing 80 years of age. I know age was included in the multivariate models, but I wondered if the authors can provide a bit more reassurance that this is accurate (i.e. does the % active decrease with age at each assessment?) and that this isn't a highly selective sample of very active individuals? Sports participation: This was the strongest predictor of later adult PA. Is there any way to tease out the type of sporting activity, or look at this in more detail to see what is driving the relationship? Sports participation duration: Also, the retrospective measure of sports participation duration – is there any validity/reliability info available for this question? I imagine being organised sport this is likely to be reasonably accurate (more so than other less structured activities) but it would be good to present some data to support this questions' psychometric properties. I also note in the discussion the comment about childhood activity being important page 16-17, lines 131-134. Participants reporting ≥ 25 years of sports participation would have been reporting back to when they were 15-34 years of age. While I appreciate the classification is greater than 25 years for those 300 men who were in this category, I do think the reference to childhood may be stretching it a bit far and would be more comfortable with the terms adolescence, youth or early adulthood here. STROBE statement – for some reason all text has been marked out with a strikethrough, seems a bit strange?
--	---

REVIEWER	Jamie McPhee Manchester Metropolitan University, UK
REVIEW RETURNED	23-May-2017

GENERAL COMMENTS	The authors have looked at self-report data on physical activity levels available over a relatively long time of follow-up in a reasonably large sample. There are some concerns over the validity of questionnaire-style self report, but the authors acknowledge this appropriately. The results are of interest and the conclusions seem appropriate. I have just a few comments to make.  1. Page 6, Lines approx 31-35. The authors say that the physical activity scores have previously been validated against objective measurements. Please provide more detail here and state clearly what the outcome was of the previous validation studies: i.e. was there a strong positive relationship with the various measurements? 2. Linked to the previous question, I can understand the validation against HR because it provides an objective measurement of intensity of activity, and I can understand the validation against "objectively measured PA" (whatever that is - please describe) because it probably validates intensity and duration of activities. The apparent "validation" against FEV1 makes less sense to me. FEV1 does not validate a questionnaire on PA. 3. I do not agree with the classification of "low recreational activity" as equal to or less than 2 hours per day on most days of the week. That seems quite a lot of recreational activity to me. Please justify and consider adding a middle group, to provide low-medium-high
--

	groups (p6, L46-50). 4. The legend for Table 1 has mixed up the "c" and "d" explanations. 5. The results suggest retiring men reported less recreational activity, but they also reported more walking. These things are discussed separately between bottom of p10 and top of p11. Have you considered that the subjects might have simply reported their activities differently at follow-up? I.e. the walking that they report to have increased is in fact a recreational activity now that they are retired?
--	--

VERSION 1 – AUTHOR RESPONSE

Response to reviewer/editorial comments

- Please revise your title so that it includes your study design (prospective cohort study?). This is the preferred format for the journal. Please also add a study design section to the abstract as per journal guidelines for research articles.

Many thanks for your comments and the opportunity to revise the paper. We have amended the title to reflect the design: "Association between physical activity levels in midlife with physical activity in old age: A 20-year tracking study *in a prospective cohort*"

We have also added a section to the abstract to clarify the design.

Reviewers' Comments to Author:

Reviewer: 1

Reviewer Name: Verity Cleland

Institution and Country: University of Tasmania, Australia

Competing Interests: None declared

1. A well and clearly-written paper addressing a research gap of interest – that is, how well does physical activity track from middle age into old age. The study uses a well-established British cohort of men recruited through general practices around 1980, so results won't be generalizable to women or non-white populations, which is appropriately acknowledged. The study found moderate levels of tracking, and provides important data that is not readily available elsewhere.

The authors mention in the Abstract that a limitation is the use of a self-report measure of physical activity which may have been prone to recall bias. This measure has been validated against heart rate and respiration, and more recently accelerometry, with acceptable properties. I think in a tracking study, as long as the recall bias is the same at each time point (would we expect people to recall differently at different time points/ages? I don't know) this should not impact on the findings in a significant way. It would only matter if people were recalling physical activity differently at different time points. The authors acknowledge this in the discussion, so I don't think it is warranted here as it detracts from the findings.

Thank you for your comment. We have removed the comment from the abstract. We agree that recall bias may have impacted our results if participants recalled differently at different time points. Plausibly, the ageing process and the associated declines in cognitive function may be an important factor that may lead to variation in recall. At 20-year follow up we asked men to compare their memory with 5 years ago. To examine if memory problems may have biased our results we performed an additional analysis excluding men whose memory had declined in the last 5 years. Kappa statistics were only fractionally higher in the sub-sample without memory problems (n=2738) (Kappas ranged from 0.24-0.27 for overall physical activity). Thus, we are confident that any variation in recall bias across time points did not significantly impact our results.

2. Measures: the authors of this paper collapsed the physical activity variables into mostly binary categories. I wonder what the justification for this was?

Kappa statistics vary in magnitude depending on how the outcome measure is categorised. To be consistent across all our measures we decided to collapse all variables into binary categories. As an additional analysis we examined tracking of overall physical activity and PA domains using 3 categories for each (see below). Kappa statistics were slightly lower across all measures but our overall conclusions remain unchanged.

Table 1. Stability of physical activity variables over time, n=3413

	Wave 1 to 2 Kappa	Wave 1 to 3 Kappa	Wave 1 to 4 Kappa
Physically activity ^a	0.24	0.21	0.22
Sport participation ^b	0.33	0.31	0.31
Recreational activity ^c	0.19	0.14	0.13
Walking ^d	0.12	0.09	0.08

^a Physical activity was categorised as low (inactive, occasional) medium (light, moderate) and high (moderately vigorous, vigorous)

^b Sport participation was categorised as none, occasional (less than once a month) and frequently (once a month or more)

^c Recreational activity was categorised as low (<4 hours at the weekend), medium (similar to 4 hours at the weekend) and high (>4 hours at the weekend)

^d Walking was categorised as low (<20 minutes/day), medium (21-60 minutes/day) and high (>60 minutes/day)

Note. Kappa statistics are presented for participants with a valid physical activity score at all four time points (n=3413). Data on walking was missing for an additional 15 participants at 12 year follow up, 3 participants at 16-year follow up and 1 participant at 20-year follow up. Data on sport participation was missing for 33 participants at 12 year follow up, 68 participants at 16 year follow up and 34 participants at 20 year follow up.

3. Results, page 8, first paragraph: It is reported that those who dropped out of the sample were older, had higher BMIs and were more likely to be inactive at baseline. It would be good if the authors could discuss the potential impact of this attrition on the findings. I acknowledge some will be included in the random effects modelling (those with two assessments) but these people again may be different to those with only one assessment or more than two assessments.

Thank you. We agree subject attrition may have biased our results and we have now discussed this further in the limitations section:

"Furthermore, men who were lost to follow up were more likely to be overweight or obese and were generally less active than men with complete data. This attrition may have led to an overestimation of physical activity levels and the strength of tracking. Physical activity may be more liable to change in men who were lost to follow up, possibly as a result of an increased risk of developing chronic health conditions (Walker et al., 1987). Random effects models which provide estimates of tracking using all available data, whilst also accounting for factors that may influence tracking strength, may have alleviated, at least in part, the bias caused by this attrition."

4. Table 1: Not clear why data for overweight/obese, smoking and occupation are absent from all but one assessment? Maybe a footnote to explain?

Thank you – The subsequent models utilise covariate data at baseline only. Thus, we only present baseline data in table 1. Although smoking is less stable, BMI and social class are also highly stable so we used only baseline measures to avoid collinearity issues.

We have added a footnote to the table to clarify:

“Data on BMI, smoking status and occupational class were utilised at baseline only”

5. Table 1: I was surprised by how active this population is, especially at the 20-year follow-up when some participants would be nearing 80 years of age. I know age was included in the multivariate models, but I wondered if the authors can provide a bit more reassurance that this is accurate (i.e. does the % active decrease with age at each assessment?) and that this isn't a highly selective sample of very active individuals?

We agree that it was quite surprising that the proportion of men who were physically active remained fairly stable throughout the follow up. However, when we stratify by age we can see that older men are generally less active than younger men from our sample (see below), apart from at 12-year follow up, which seems to coincide with typical retirement age for the older men.

Table 2. Proportion physically active stratified by age

	Baseline		12 year		16 year		20 year	
	Younger ^a	Older ^b	Younger	Older	Younger	Older	Younger	Older
Physically active [†] (% , n)	69.1 (1180)	63.1 (1077)	68.8 (1175)	73.1 (1247)	65.1 (1111)	62.3 (1062)	70.4 (1202)	63.4 (1082)

^a Younger men were aged between 38 and 47 years at baseline

^b Older men were aged between 48 and 60 years at baseline

[†] Physically active was classified as reporting at least light activity

Also, when we extended the follow up to 30 years (when men were aged 70-89), we did observe a substantial decline in the number of men who are classed as active (data not shown). This is consistent with cross-sectional data from the Health Survey for England (Scholes et al., 2012), which presented similar proportions of men meeting physical activity recommendations at ages 55 to 64 and at ages 65 to 74 years, followed by a decline in the 75+ year olds. We chose not to extend the follow up period to 30 years in this paper in order to maximise our sample size. Finally, looking at the domains of activity we see that the high proportion of men classified as active is being driven by increases in walking. This is consistent with other studies that have reported a shift from structured activities to walking in older age groups.

6. Sports participation: This was the strongest predictor of later adult PA. Is there any way to tease out the type of sporting activity, or look at this in more detail to see what is driving the relationship?

Thank you for your suggestion. We think this is a good idea, but unfortunately we do not have the data available at present to investigate this fully. We are currently seeking funding to investigate this in a future study when we can robustly determine the types of sport at each wave.

7. Sports participation duration: Also, the retrospective measure of sports participation duration – is there any validity/reliability info available for this question? I imagine being organised sport this is likely to be reasonably accurate (more so than other less structured activities) but it would be good to present some data to support this questions' psychometric properties. I also note in the discussion the comment about childhood activity being important page 16-17, lines 131-134. Participants reporting >=25 years of sports participation would have been reporting back to when they were 15-34 years of age. While I appreciate the classification is greater than 25 years for those 300 men who were in this

category, I do think the reference to childhood may be stretching it a bit far and would be more comfortable with the terms adolescence, youth or early adulthood here.

Thank you for your comment. Unfortunately we have no means to validate this question but similar questions have been used in similar age groups asking for recall over longer periods (Dohle et al., 2013). We have added some text to highlight this limitation:

"Despite this, we are unable to validate responses to the question on duration of sport participation, although studies in men of a comparable age have used similar questions with longer recall periods."

We have also amended the sentence regarding the reference to childhood:

"Engagement in sport by *early adulthood* may be crucial for establishing a lifelong habit for sport participation and for developing *important* motor skills."

8. STROBE statement – for some reason all text has been marked out with a strikethrough, seems a bit strange?

Thank you – we have removed the strikethroughs and added the page numbers to the statement.

Reviewer: 2

Reviewer Name: Jamie McPhee

Institution and Country: Manchester Metropolitan University, UK

Competing Interests: None declared

The authors have looked at self-report data on physical activity levels available over a relatively long time of follow-up in a reasonably large sample. There are some concerns over the validity of questionnaire-style self report, but the authors acknowledge this appropriately. The results are of interest and the conclusions seem appropriate. I have just a few comments to make.

1. Page 6, Lines approx 31-35. The authors say that the physical activity scores have previously been validated against objective measurements. Please provide more detail here and state clearly what the outcome was of the previous validation studies: i.e. was there a strong positive relationship with the various measurements?

Thank you. We can confirm there were strong associations between our PA scores and measures of fitness, heart rate and accelerometer measured physical activity. We have added information on the relationships with the respective measures in the validation studies:

"These PA scores have previously been validated against heart rate, forced expiratory volume in 1 second (FEV1)¹ and accelerometer measured PA². Results from the validation studies revealed a strong inverse relationship between PA score and heart rate and a strong positive association with FEV1 and with accelerometer measured moderate-to-vigorous PA ($r=0.49, p < 0.001$)^{1, 2}."

2. Linked to the previous question, I can understand the validation against HR because it provides an objective measurement of intensity of activity, and I can understand the validation against "objectively measured PA" (whatever that is - please describe) because it probably validates intensity and duration of activities. The apparent "validation" against FEV1 makes less sense to me. FEV1 does not validate a questionnaire on PA.

Thank you – we have added a few words to explain that our questionnaire was validated against accelerometer-measured physical activity levels.

"...more recently against accelerometer measured PA"

Fitness levels are determined by habitual activity levels therefore we would expect to see an association between our measure of physical activity and measures of fitness (i.e. FEV1). We agree other measures of fitness would be more appropriate (e.g. VO² max), but FEV1 should also be correlated with physical activity levels. Providing evidence that our measure of physical activity is associated with variables that, in theory, physical activity should be, demonstrates convergent validity, an important subcategory of construct validity.

3. I do not agree with the classification of "low recreational activity" as equal to or less than 2 hours per day on most days of the week. That seems quite a lot of recreational activity to me. Please justify and consider adding a middle group, to provide low-medium-high groups (p6, L46-50).

Thank you for your comment. This relates to the second comment by reviewer 1. Kappa statistics vary in magnitude depending on how we categorise outcome measures. To be consistent across all our measures we decided to collapse all variables into binary categories. We have repeated part of the analyses with three categories for overall physical activity and each domain (see table above). Classifying men as low (<4 hours at the weekend), medium (similar to 4 hours at the weekend) or high (>4 hours at the weekend) recreational activity, we observed slightly lower levels of tracking compared to the binary categorisation and comparable declines (from 52% at baseline to 38% at 20-year follow up). Overall our conclusions do not change with the 3 group categorisation.

4. The legend for Table 1 has mixed up the "c" and "d" explanations.

Thank you. We have corrected the footnote.

5. The results suggest retiring men reported less recreational activity, but they also reported more walking. These things are discussed separately between bottom of p10 and top of p11. Have you considered that the subjects might have simply reported their activities differently at follow-up? I.e. the walking that they report to have increased is in fact a recreational activity now that they are retired?

Our measure of time spent walking includes all forms of walking (travel and recreational). So at all time points our walking measure is an indicator of overall walking time regardless of whether that was derived from travel or recreational walking. With regards to our measure of recreational activity, we agree that the components of recreational activity may have changed over time to include more recreational walking. However, including recreational walking in this domain may only have masked an even steeper decline in other recreational activities, if indeed recreational walking did increase. We have added a sentence to discuss this:

"The measure of recreational activity consisted of several activities including recreational walking. Given that recreational activity decreased, increases in walking may have masked an even steeper decline in other recreational activities."

VERSION 2 – REVIEW

REVIEWER	Verity Cleland University of Tasmania, Australia
REVIEW RETURNED	11-Jul-2017

GENERAL COMMENTS	The authors have been thorough in their responses and I am satisfied with the manuscript with the exception of two minor points: VC2. Can the authors please add this justification to the manuscript? VC5. Can the authors please add some of this to the Discussion?
--

REVIEWER	Jamie McPhee Manchester Met University, UK
REVIEW RETURNED	30-Jun-2017

GENERAL COMMENTS	I thank the authors for their responses to the questions I raised. I am happy with the responses, except for one. I still do not fully understand the validation of the physical activity classification, and I cannot accept that FEV1 can be used to validate a physical activity questionnaire or classify physical activity levels. The authors state "The total PA score was then used to classify activity levels as inactive, occasional, light, moderate, moderately vigorous or vigorous. These PA scores have previously been validated against heart rate, forced expiratory volume in 1 second (FEV1) 19 and accelerometer-measured PA 20. Results from the validation studies revealed a strong inverse relationship between PA and heart rate and a strong positive association with FEV1 and accelerometer-measured moderate-to-vigorous PA ($r=0.49$, $p < 0.001$)^{19, 20}."  - Which of the various validators does the "$r=0.49$, $p<0.001$" represent? You also need to provide the r-values for the other validators. - Specifically, how was the PA classification validated against HR? e.g. was HR measured continuously over several days and results compared across the PA groups? - FEV1 will be lower in people with cardiorespiratory conditions, such as COPD, and these people are likely to have low activity levels. I accept that. However, FEV1 does not distinguish between healthy older people with vastly different physical activity levels, physical fitness and mobility status. Here are two publications that show FEV1 is 1) only weakly related to mobility amongst older healthy adults, and 2) only slightly higher in very athletic compared to non athletic older people, and not related to physical performance or training history. 1. Silanpaa et al (Age (Dordr). 2014 Aug; 36(4): 9667.). https://www.ncbi.nlm.nih.gov/pmc/articles/PMC4150884/ 2. Degens et al (Age (Dordr). 2013, 35(3), pp 1007–1015). https://link.springer.com/article/10.1007%2Fs11357-012-9409-7?LI=true My recommendation is to remove the mention of FEV1 as a validation of the physical activity classification.
---

VERSION 2 – AUTHOR RESPONSE

Reviewer: 1

Reviewer Name: Verity Cleland

Institution and Country: University of Tasmania, Australia Competing Interests: None declared.

The authors have been thorough in their responses and I am satisfied with the manuscript with the exception of two minor points:

VC2. Can the authors please add this justification to the manuscript?

Thank you. We have added the following to justify using binary variables:

“Kappa statistics vary in magnitude depending on how the outcome measure is categorised. To be consistent across all our measures we decided to perform analyses using binary variables.”

VC5. Can the authors please add some of this to the Discussion?

We agree this point is important to include, thus we have added the following statement to our results section:

“The proportion of men classified as active declined more rapidly thereafter, with 57.3% of men classified as active at 30-year follow up (data not shown).”

And commented on this further in the discussion:

“Although the proportion of men categorized as active fluctuated over the 20-year follow up period, we did not observe the decline over time that one might expect with the advancing age of the sample. However, when we extended our follow up to 30 years when men were aged 70-89, a notable decline was observed. This is consistent with cross-sectional data from the Health Survey for England (Scholes et al., 2012), which presented similar proportions of men meeting physical activity recommendations at ages 55 to 64 and at ages 65 to 74 years, followed by a decline in men >75 years old.”

Reviewer: 2

Reviewer Name: Jamie McPhee

Institution and Country: Manchester Met University, UK Competing Interests: None declared

I thank the authors for their responses to the questions I raised. I am happy with the responses, except for one. I still do not fully understand the validation of the physical activity classification, and I cannot accept that FEV1 can be used to validate a physical activity questionnaire or classify physical activity levels.

The authors state "The total PA score was then used to classify activity levels as inactive, occasional, light, moderate, moderately vigorous or vigorous. These PA scores have previously been validated against heart rate, forced expiratory volume in 1 second (FEV1) 19 and accelerometer-measured PA 20. Results from the validation studies revealed a strong inverse relationship between PA and heart rate and a strong positive association with FEV1 and accelerometer-measured moderate-to-vigorous PA (r=0.49, p < 0.001)19, 20."

We thank the reviewer for their comment and have made adjustments to the text to clarify the results from the validation studies and to remove the reference to FEV1.

- Which of the various validators does the "r=0.49, p<0.001" represent? You also need to provide the r-values for the other validators.

This refers to the accelerometer measured PA, and we have attempted to clarify this in the text. We report the p value for the association between PA scores and resting heart rate – r values were not reported in the original investigation.

“Results from the validation studies revealed a strong inverse relationship between PA and electrocardiogram-measured resting heart rate (p<0.001) and a strong positive association with accelerometer-measured moderate-to-vigorous PA (r=0.49, p < 0.001)^{19, 20}”

- Specifically, how was the PA classification validated against HR? e.g. was HR measured continuously over several days and results compared across the PA groups?

Resting heart rate was measured via electrocardiogram at the baseline examination by a research nurse and was then compared with PA scores at baseline. We have included more information on the methods in the sentence above.

- FEV1 will be lower in people with cardiorespiratory conditions, such as COPD, and these people are likely to have low activity levels. I accept that. However, FEV1 does not distinguish between healthy older people with vastly different physical activity levels, physical fitness and mobility status. Here are two publications that show FEV1 is 1) only weakly related to mobility amongst older healthy adults, and 2) only slightly higher in very athletic compared to non athletic older people, and not related to physical performance or training history.

1. Silanpaa et al (Age (Dordr). 2014 Aug; 36(4): 9667.).
<https://www.ncbi.nlm.nih.gov/pmc/articles/PMC4150884/>

2. Degens et al (Age (Dordr). 2013, 35(3), pp 1007–1015).
<https://link.springer.com/article/10.1007%2Fs11357-012-9409-7?LI=true>

My recommendation is to remove the mention of FEV1 as a validation of the physical activity classification.

We are in agreement that FEV1 is not the most appropriate measure for providing validity evidence for physical activity in older adults and, therefore, we have removed references to FEV1.

By way of background, the cohort study uses a physical activity questionnaire which has been in use for many decades (since 1978), the questionnaire is kept the same at subsequent waves of the study in order to permit comparisons over time. Prior to initial use in 1978 the questionnaire was not validated against measures that we might choose if we were starting today. However, we have used the best data that we have available from our cohort study to assess whether the results from the questionnaire are related to physical measures that we would a priori expect them to be related to. Hence our comparisons with ECG resting heart rate, FEV1 and more recently, accelerometer measured physical activity- these are the most relevant data available. We agree with the reviewer that these are not the measures that we would choose to validate a PA score against if we were

starting the project today in a younger population and with a different clinical setting. The study is a national cohort study which means that men are measured in 24 towns distributed all across the UK, necessitating us to travel long distances and use portable equipment. We do not have the means to put the participants through gold standard exercise testing to calculate individual level energy expenditure. Furthermore, given that the men are now aged on average 88 (range 80-100) years, something like a submaximal treadmill test would be highly problematic among this population. However, we can reassure the reviewer that the score has construct validity and content validity (see results from the two validation papers quoted above). It has also been accepted and very widely

published in high impact factor journals including for example the following: (Wannamethee et al., Lancet, 1998. Changes in physical activity, mortality, and incidence of coronary heart disease in older men; Wannamethee et al., Circulation, 2000. Physical activity and mortality in older men with diagnosed coronary heart disease; Wannamethee et al., Heart, 1999. Role of risk factors for major coronary heart disease events with increasing length of follow up; Jefferis et al., Diabetes Care, 2012. Longitudinal associations between changes in physical activity and onset of type 2 diabetes in older British men: the influence of adiposity). As a result of the most recent concurrent validation against accelerometer measured activity, we have extra reason to be confident that the questionnaire categorises participants according to their level of activity and according to their type of activity.

VERSION 3 – REVIEW

REVIEWER	Jamie McPhee Manchester Metropolitan University, UK
REVIEW RETURNED	16-Jul-2017

GENERAL COMMENTS	The authors have responded to my final concern appropriately. I am grateful for the additional background information that they provided. I fully understand the limitations of having to stick with the original methodology of the 1970s in order to maintain consistency over the follow-up years. My point is that stating that PA was validated against FEV1 could cause some people to question the validity of the PA questionnaire. This would be a shame because it is unnecessary and avoidable. I think the authors made a good decision to remove the mention of FEV1 from the manuscript.
--